# Mediating Effect of Quality of Sleep Moderated by Meaning in Life on the Relationship between Hwabyung and Suicidal Ideation in Middle-Aged Korean Women

**DOI:** 10.3390/bs13060509

**Published:** 2023-06-19

**Authors:** Goo-Churl Jeong, Jae-Sun An, Sun-Hwa Shin

**Affiliations:** 1Department of Counselling Psychology, Sahmyook University, Seoul 01795, Republic of Korea; gcjeong@syu.ac.kr (G.-C.J.); lucky603@syu.ac.kr (J.-S.A.); 2Nursing Department, College of Nursing, Sahmyook University, Seoul 01795, Republic of Korea

**Keywords:** middle-aged women, Hwabyung, quality of sleep, meaning in life, suicidal ideation

## Abstract

The purpose of this study was to determine the role of the quality of sleep and meaning in life in the process by which Hwabyung symptoms affect suicidal ideation in middle-aged Korean women. A total of 265 women aged 40–65 years were enrolled in an online survey. The study variables were measured using the Hwabyung, quality of sleep, meaning in life, and suicidal ideation scales. The data were analyzed using the PROCESS Procedure for SPSS Release 3.5 (Model 14) program with a 95% bias-corrected bootstrap confidence interval. Hwabyung symptoms in middle-aged women had a significant direct effect on suicidal ideation, and an indirect effect through the quality of sleep was also statistically significant. Meaning in life was found to significantly moderate the indirect effect of Hwabyung on suicidal ideation through the quality of sleep. In other words, the greater the meaning in life, the weaker the effect of Hwabyung on suicidal ideation through the quality of sleep. The Hwabyung of middle-aged women caused a psychological crisis and was a great threat to physical health by lowering the quality of sleep. The low quality of sleep and the increase in suicidal ideation due to Hwabyung pose a great threat to the survival of middle-aged women. It was found that it is very important to find meaning in one’s life as an effective way to reduce suicidal ideation in middle-aged women.

## 1. Introduction

As of 2019, the average suicide rate in Organization for Economic Cooperation and Development (OECD) member countries was 11 suicide attempts per month, whereas, in Korea, it was 24.6, which is 2.2 times higher (38.0%) [1]. Suicidal ideation is a preliminary variable that leads to suicide, and access to suicide data can be carried out through research on suicidal ideation. The stress vulnerability model explains the cause of suicide as an interaction between an individual’s psychological characteristics and events occurring in his or her environment. Psychological vulnerability can be a major cause of increased suicidal ideation. Depression is the main cause of suicidal ideation in middle-aged adults in Korea, and for middle-aged men, income level affected suicidal ideation, while for middle-aged women, psychological factors, such as having a spouse, self-esteem, and family conflict, did so [2]. A study on suicidal ideation in middle-aged men, with the highest mortality rate caused by honorary retirement and depression, was attempted during a financial crisis [3,4]. However, studies on suicidal ideation among middle-aged women are rare.

Middle-aged women experience various physical and psychological symptoms due to hormonal decline and imbalance as they physiologically encounter menopause [5,6,7]. Menopause is characterized by a decrease in the production of female sex hormones by the ovaries, resulting in a variety of symptoms, including vascular changes (hot flashes), genitourinary changes, emotional changes (depression, anxiety, nervousness, difficulty concentrating, etc.), and musculoskeletal changes (osteoporosis) [5,6]. These physical and psychological changes and long-term stress can often lead to development of the Hwabyung. It is particularly noteworthy that most women who experience Hwabyung are middle-aged women in their 40s and 50s [8,9,10]. Psychological vulnerability is the main cause of suicidal ideation in middle-aged Korean women, and “Hwabyung”, a culture-bound illness caused by the characteristics of Korea’s unique culture, can be considered such a vulnerability. Hwabyung is a kind of mental disorder that arises due to the long-term accumulation of injustice and resentment experienced in Korea’s traditional Confucian cultural environment, that is, patriarchal and authoritative with the suppression of emotional expression [11]. In particular, middle-aged Korean women experience a “Hwabyung” that explodes because they are unable to express their regret about the situation they are in while fulfilling strict gender roles in their relationships with their families, children, and parents-in-law [10,12]. The Hwabyung symptoms experienced during a vulnerable period (such as the onset of menopause place) are a serious burden on individuals and their families, which in turn causes them to experience significant psychological difficulties, such as low self-esteem and increased self-criticism [10,13]. Previous studies on Hwabyung have mainly dealt with the causes and symptoms of Hwabyung, while others have focused on the personality characteristics, defense mechanisms, and coping methods of Hwabyung patients [9,14]. Most studies that attempted a therapeutic approach to Hwabyung confirmed the effect of intervention using an oriental medicine approach [15,16]. The effect of such an intervention was verified through a psychological approach using counseling and meditation [17,18,19].

Physical symptoms during menopause include facial redness, sweating, sleep disorders, and fatigue, which become more severe as women enter late menopause [20]. Insomnia is a representative physical symptom that appears in the early stages of Hwabyung, and sleep time and quality are related to Hwabyung symptoms [21,22,23]. The Hwabyung experienced by middle-aged women before and after menopause is a symptom that should be treated seriously because it can affect extreme thoughts, such as suicide [24]. When looking at the results of previous studies that showed that decreased sleep time increased suicidal ideation [25,26,27], it is evident that a continuous decrease in sleep time can hinder healthy daily life factors. A correlation between Hwabyung, quality of sleep, and suicidal ideation is evident in such literature. Hwabyung tends to appear more often in women after middle age, and if it is not properly managed, it can persist into old age and affect the suicide risk in older adults [22]. However, middle-aged women in Korea continue to suffer from Hwabyung, and they are known to experience serious physical pain and sleep disorders due to it [23]. Therefore, appropriate interventions are needed so that the physical pain of Hwabyung symptoms and sleep disturbance, which are culturally linked syndromes experienced by middle-aged Korean women, do not increase suicidal ideation.

Those who experience suicidal ideation often feel no reason to live, so it is especially important to instill more meaning into the lives of patients and help them find reasons to live to reduce suicidal ideation. Meaning in life is a powerful force in strengthening positive and mitigating negative outlooks [28]. Previous studies have reported that people with more meaning in life experience greater satisfaction or happiness owing to increased positive emotions [29]. As such, the experience of meaning in life is a positive psychological mechanism that can reduce mental health problems such as Hwabyung and complications such as suicidal ideation. Meaning in life (or the reason for life) can act as a moderating variable that weakens the effects of depression or stress on suicidal ideation [30,31]. The meaning in life buffers pain when experiencing misfortune or negative events and plays a role in changing not only the attitude toward pain but also the attitude toward life itself [28,32]. If middle-aged women actively seek meaning in their lives through positive interactions, not negative interactions with the sociocultural environment, the hope is that the rate of suicidal ideation caused by Hwabyung symptoms will decrease.

Based on the literature, Hwabyung has a significant impact on suicidal ideation [22]. Hwabyung is negatively correlated with sleep time [22,23], and the longer the sleep time and the higher the quality of sleep, the lower the suicidal ideation [22,25,27,33]. Thus, we hypothesized that the quality of sleep would play a role as a parameter in the process by which Hwabyung symptoms affect suicidal ideation. In addition, meaning in life served as a moderating variable in the relationship between depression and suicidal ideation [30] and between life stress and suicidal ideation [31]. Based on previous findings, this study examined the mediating effects of the quality of sleep and meaning in life in the process by which Hwabyung symptoms affect suicidal ideation in middle-aged women. By confirming the relationship between the study factors, we intend to provide basic data for developing effective interventions to reduce suicidal ideation and promote mental health among middle-aged Korean women.

## 2. Materials and Methods

### 2.1. Study Design and Participants

This was a cross-sectional study using an online survey targeting middle-aged women who met the inclusion criteria through various online communities. The study participants were married, middle-aged women aged 40–65 years. Currently unmarried or foreign-born women were excluded from the study.

A total of 265 responses were used in the final analysis. Looking at the characteristics of the study participants, 128 women (48.3%) were in their 40s, and 137 women (51.7%) were in their 50s or older. Looking at the characteristics of the study participants, 128 women (48.3%) were in their 40s, 107 women (40.4%) were in their 50s, and 30 women (11.3%) were in their 60s. Of the study participants, 111 women (41.9%) had full-time jobs, 55 women (20.8%) had part-time jobs, and 99 women (37.4%) were housewives. Regarding educational background, 200 women (75.4%) had a college degree or higher, and 65 women (24.6%) had a high school education or less.

### 2.2. Survey Conduct

This study was approved by the Institutional Review Board of Sahmyook University (IRB no. SYU 2022-02-013) on 4 March 2022. Data were collected through an online survey conducted in April 2022. The researchers used email and social media to conduct convenience sampling, focusing on people in their immediate neighborhood. The online survey was only conducted if the participants read the researcher’s instructions before taking the survey and voluntarily agreed to participate. In the researcher’s explanatory note, the purpose of this study, limitations on the use of collected data, and the fact that there was no penalty for not participating in the questionnaire or giving up while participating were specified. All participants signed an informed consent form via the online questionnaire.

Four scales were used to measure Hwabyung, quality of sleep, meaning in life, and suicidal ideation in the participants.

Hwabyung was measured using the Hwabyung Symptom Scale (HSS) developed by Kwon et al. [14]. The sub-factors of HSS were composed of emotional symptoms (11 items) and somatic symptoms (6 items). The HSS consists of 15 questions evaluated on a 5-point Likert scale. The higher the score, the more severe the symptoms of Hwabyung. In this study, Cronbach’s α value was 0.93.

Quality of sleep was measured using the Subjective Sleep Scale (SSS) developed by Snyder-Halpern and Verran [34,35]. The scale consists of 15 questions evaluated on a 4-point Likert scale. The higher the score, the better the quality of sleep. In this study, Cronbach’s α value was 0.90.

Meaning in life was measured using the Meaning in Life Scale (MLS) developed by Steger et al. [36,37]. The MLS consists of 10 questions evaluated on a 5-point Likert scale. The higher the score, the higher the respondent’s meaning in life. In this study, Cronbach’s α value was 0.93.

Suicidal ideation was measured using the Suicidal Ideation Scale (SIS) developed by Reynold [38,39]. The SIS consists of 14 items evaluated on a 7-point Likert scale. The higher the score, the higher the degree of suicidal ideation. In this study, Cronbach’s α was 0.95.

### 2.3. Statistical Methods

The collected data were analyzed using the IBM SPSS program version 25.0 (IBM Corp., Armonk, NY, USA). The mean and standard deviation of the major research variables were calculated, and a frequency analysis was performed for the general characteristics. The reliability of the scale is presented as an internal consistency coefficient (Cronbach’s α). Correlations between the study variables were analyzed using Pearson’s product correlation analysis. The moderated mediating effect analysis was performed using the PROCESS Procedure for SPSS Release 3.5 (Model 14) program based on regression analysis, and the Johnson–Neyman significant domain of the moderating effect was analyzed and is presented.

## 3. Results

### 3.1. Demographic Characteristics and Suicidal Ideation of Participants

Differences in Hwabyung, the quality of sleep, meaning in life, and suicidal ideation by general characteristics were analyzed by ANOVA. The results showed that women in their 60s had significantly more suicidal thoughts than women in their 30s (*F* = 4.41, *p* = 0.013). In terms of education, women with a college degree or higher had significantly higher meaning in life than women with a high school diploma or less (*F* = 8.83, *p* = 0.003). There were no statistical differences in the study variables by employment type or monthly income. As no confounding variables were found in the general characteristics, we did not control for them in the regression models.

The results of the analysis of some of the suicidal ideation questions for middle-aged women are illustrated in Figure 1. Middle-aged women answered “yes” to “I thought about having a big accident”, with the highest rate of 60.4%. In addition, a high proportion of middle-aged women thought about how to attempt suicide (47.2%) and when to attempt suicide (33.2%). On the other hand, only 12.5% of the participants reported their suicidal ideation to others, indicating less communication with those around them.

### 3.2. Result of Correlation Analysis between Research Variables

We calculated the averages, standard deviations, and correlation coefficients of Hwabyung, quality of sleep, meaning in life, and suicidal ideation (see Table 1). Hwabyung symptoms in middle-aged women were significantly negatively correlated with the quality of sleep (r = −0.46, *p* < 0.001) and meaning in life (r = −0.34, *p* < 0.001). There was a significant positive correlation between Hwabyung and suicidal ideation (r = 0.49, *p* < 0.001). The quality of sleep had a weak positive correlation with meaning in life (r = 0.19, *p* < 0.01) and a significant negative correlation with suicidal ideation (r = −0.34, *p* < 0.001). There was a significant negative correlation between meaning in life and suicidal ideation (r = −0.28, *p* < 0.001).

### 3.3. Moderated Mediating Effect of Quality of Sleep through Meaning in Life

The mediating effect of the quality of sleep moderated by meaning in life on the relationship between Hwabyung and suicidal ideation in middle-aged women is presented in Table 2. For ease of interpretation, we included the mean-centering variables of Hwabyung, quality of sleep, and meaning in life in the regression model. As a result of the regression analysis, the Durbin–Watson value was 1.98, assuming the independence of the error term. The Variance Inflation Factor (VIF) values of the predictors for the dependent variable were 1.38 or less, indicating that there was no multicollinearity problem. The amount of explanation in the regression model was statistically significant at 28.2% (R^2^ = 0.282, *F* = 25.52, *p* < 0.001).

A model of the mediating effect of the quality of sleep moderated by meaning in life on the effect of Hwabyung on suicidal ideation is presented in Figure 2 with regression coefficients.

To better understand the moderating effect of meaning in life on the relationship between the quality of sleep and suicidal ideation in middle-aged women, the conditional effect of the quality of sleep according to the level of meaning in life is presented in Figure 3a. As shown in Figure 3a, the negative effect of the quality of sleep on suicidal ideation weakened as meaning in life increased.

For the moderating effect, the Johnson–Neyman significance region was calculated and is presented in Figure 3b. For ease of interpretation, meaning in life was plotted using the raw score before mean centering. In this study, the increase in suicidal ideation due to poor quality of sleep was statistically significant only in middle-aged women (51.7%) with low meaning in life.

We analyzed the conditional mediating effect of the quality of sleep moderated by meaning in life on the relationship between Hwabyung and suicidal ideation in middle-aged women using the bootstrapping method (see Table 3). At the lower level of meaning in life (M − 1SD), the indirect effect of Hwabyung on suicidal ideation through the quality of sleep was statistically significant (B = 0.11, 95% CI [0.03, 0.20]); however, it was not statistically significant at the upper level (M + 1SD). In summary, Hwabyung had a significant direct effect on suicidal ideation, and the indirect effect through the quality of sleep was statistically significant only at a low level of meaning in life.

## 4. Discussion

This study aimed to provide basic data for developing effective interventions to improve the mental health of middle-aged women in Korea by examining the relationship between Hwabyung, the quality of sleep, and suicidal ideation. In particular, it was confirmed that meaning in life plays a key role in reducing suicidal ideation in middle-aged women by confirming that the mediating effect of the quality of sleep was controlled by meaning in life.

The symptoms of Hwabyung in middle-aged women had a direct effect on suicidal ideation. In previous studies, Hwabyung in middle-aged women before and after menopause was associated with suicide [24]. In addition, Hwabyung symptoms in older populations have been reported to have a direct effect on suicidal ideation [22], which is consistent with the results of this study. The symptoms of Hwabyung are the accumulation of resentment and anger caused by long-term stress [13], which negatively affects daily life and quality of life [40]. In previous studies, the meaning of the experience of middle-aged Korean women’s Hwabyung were “being alone”, “continuous suffering”, and “ruining children’s farming” [41]. In Korea’s patriarchal and authoritative atmosphere, when the resentment of suppressing one’s, emotional expression has lasted for a long time, it can cause an explosion, like a Hwabyung under fire [10,11,12]. In this way, “unfairness” is a factor that hinders mental health in Korea, with manifestations ranging from Hwabyung to suicide [12]. As such, Hwabyung is related to a sociocultural environment that causes injustice and anger, and it has been reported that Hwabyung patients have stronger negative emotions of depression, anxiety, and anger than healthy people [17]. This means that people experiencing Hwabyung symptoms may struggle to cope with, control, or accept negative emotions in their situation. In addition, it suggests that attention should be paid to prevent such psychological vulnerability from leading to an increased risk of suicide. Therefore, suicide risk assessments for middle-aged Korean women should be conducted regularly, and suicide prevention programs should be provided by a mental health welfare center in the community based on the assessment results. In the case of middle-aged women in Korea, understanding the sociocultural characteristics of Korea and carrying out a therapeutic approach should be prioritized above all else.

The symptoms of Hwabyung in middle-aged women lead to a reduced quality of sleep. Insomnia is a representative early symptom of Hwabyung, and in previous studies, various oriental medical interventions were applied to improve sleep disorders (e.g., insomnia) in Hwabyung patients [15,16,21]. Previous studies have reported that psychological factors, such as depression or anxiety, can cause sleep disorders [42]. It was also reported that the shorter the sleep time, the stronger the quality of sleep’s relationship with Hwabyung [23]. In this study, the worse the Hwabyung symptoms, the lower the quality of sleep, confirming that Hwabyung symptoms affect physical health by affecting sleep. The symptoms of Hwabyung include increased body heat, chest tightness, headaches, and digestive problems [43] and a combination of other physical and psychological symptoms [13]. A previous study on Hwabyung reported that disease history, economic level, marital status, academic background, and failure to get a good night’s sleep led to Hwabyung [23]. If negative external environments and stressful situations continue, negative emotions (e.g., anger) may accumulate, causing various physical symptoms. Sleep is one of our most basic physiological needs [44]. Therefore, it is necessary to provide educational interventions to improve sleep hygiene and promote sleep health in middle-aged women experiencing Hwabyung symptoms.

The decrease in the quality of sleep among middle-aged women serves as a factor in increasing the degree of suicidal ideation. Previous studies reported a negative correlation in which suicidal ideation increased as sleep time and quality decreased [22,25,26,27]. The present study also showed that suicidal ideation increases as the quality of sleep decreases. In another study, adults over the age of 50 years with depressive symptoms had a higher chance of suicide as their quality of sleep deteriorated [33]. Sleep is a physiological mechanism that reduces physical fatigue and controls negative emotions experienced during the day through information processing and hormone action in the brain. Maintaining normal sleep is a very important part of maintaining human health [45]. In general, both short and excessively long sleep hours fail to achieve sufficient physical recovery and act as factors that increase depression or suicidal ideation [26,46]. In particular, in middle-aged women, the prevalence of insomnia or sleep disorders increases with the onset of menopause [47,48]. These results show that middle-aged women are more likely to experience suicidal ideation if they do not have a sufficient quality of sleep. Therefore, it is necessary to actively manage the quality of sleep in middle-aged women.

The quality of sleep had a significant mediating effect on the relationship between Hwabyung and suicidal ideation among middle-aged women. In other words, Hwabyung symptoms decreased the quality of sleep and increased suicidal ideation. A direct comparison was difficult because there were no previous studies that analyzed the mediating model used in this study. However, in a previous study, Hwabyung had a direct effect on suicidal ideation, and a significant correlation between sleep duration and suicidal ideation was found [22], supporting the results of this study. Hwabyung is a disorder caused by the accumulation of anger-inducing stress and characterized by symptoms of depression, anxiety, and somatization [13]. In this study, 60.4% of the middle-aged women had thoughts about suicide. In addition, 47.2% of the middle-aged women thought about a specific way to attempt suicide. Since the act of “suicide” is socially and culturally taboo, these women may contemplate attempting suicide in a way that reduces moral consequences. However, the result that 47.2%, or close to half, of the middle-aged women in this study thought about a specific suicide method is a dangerous sign. When suicidal ideation materializes, they can transition to suicidal behaviors; therefore, urgent measures are needed to counteract suicidal ideation in this population.

The moderating effect of meaning in life was found to be significant in the process by which the quality of sleep affects suicidal ideation in middle-aged women. In previous studies, meaning in life had a significant moderating effect on the relationship between stress and suicidal ideation [31,49], depression’s effect on suicidal ideation [30], and stress’ effect on suicidal ideation [31]. Although this study’s research variables are not consistent with previous studies, it was found that meaning in life weakens the influence of individual stress and negative emotions, such as depression, on suicidal ideation. In this study, middle-aged women with greater meaning in life did not show an increase in suicidal ideation due to Hwabyung symptoms and low quality of sleep. This suggests that meaning in life is a strong protective factor that protects against the current anger, anguish, and physical pain that led to suicidal ideation. In previous studies, meaning in life was closely related to an individual’s subjective well-being; in particular, middle-aged adults showed a high correlation between meaning in life and depression [50]. Meaning in life is a driving force that determines the direction one takes in life, and if the meaning in life is lost, the initiative to live may also be lost [51]. In addition, meaning in life has the power to reinforce positive emotions [28]. Factors affecting meaning in the lives of middle-aged Korean adults include life satisfaction, religion, coping behavior, and self-esteem [51]. Middle-aged women in Korea experience life crises due to physical aging, menopause, and “empty nest syndrome” arising from the loss of the parental role [52]. Since such crises can cause life to lose its meaning, it is necessary to actively seek meaning in one’s life when experiencing a life crisis. Various studies have been conducted to improve meaning in life. Previous studies have applied programs such as meaning enhancement [53,54,55] and gratitude promotion [56] to promote positive self-understanding, which has been effective in enhancing meaning in life. When mindfulness meditation was applied to middle-aged women, their levels of Hwabyung symptoms, depression, anxiety, and stress decreased [19]. Thus, a method to cope with Hwabyung through psychological meditation was verified. To improve the psychological health of middle-aged women, it is necessary to develop support programs that promote an understanding of meaning in life and reduce extreme measures taken in response to a loss of life purpose.

The limitations of this study are as follows: The data were collected through online community sampling, which may not be representative of larger populations. Therefore, future studies should expand generalization through representative sampling and nationwide population proportional allocation. As research on Hwabyung is limited to the Korean culture, these findings cannot be applied to other cultures. However, we suggest conceptual analysis and comparative research be conducted to determine whether the findings can be applied in relation to psychological pain caused by anger or resentment (i.e., features of Hwabyung) seen in other cultures. Despite these limitations, this study took a concrete approach to Hwabyung experienced by middle-aged women due to the sociocultural characteristics of Korea and analyzed the relationship between the quality of sleep and suicidal ideation and the moderating effect of meaning in life. With these findings, we present an alternative foundation that may reduce suicidal ideation among middle-aged women in Korea.

## 5. Conclusions

This study confirmed that the more severe the symptoms of Hwabyung in middle-aged Korean women, the lower their quality of sleep and the greater their suicidal ideation. Hwabyung, often experienced by middle-aged women in Korea, is a culture-bound illness that causes psychological and physical difficulties in daily life. Hwabyung threatens not only physical health, which lowers the quality of sleep, but also mental health, leading to extreme thoughts of suicide. To improve the health of middle-aged women in Korea, future studies should endeavor to further understand Korea’s cultural background and examine the duration of Hwabyung symptoms. Lifestyle improvement education is needed to maintain healthy sleep hygiene, as improved quality of sleep can reduce suicidal ideation in middle-aged Korean women. In addition, interventions and strategies to identify the purpose and value of life should be implemented to support the experience of meaning in life. At the national level, measures should be taken to appropriately cope with the mental health crises experienced by middle-aged women in Korea, and efforts should be made to improve the sociocultural environment and awareness surrounding psychological health so that Hwabyung does not become a common experience.

## Figures and Tables

**Figure 1 behavsci-13-00509-f001:**
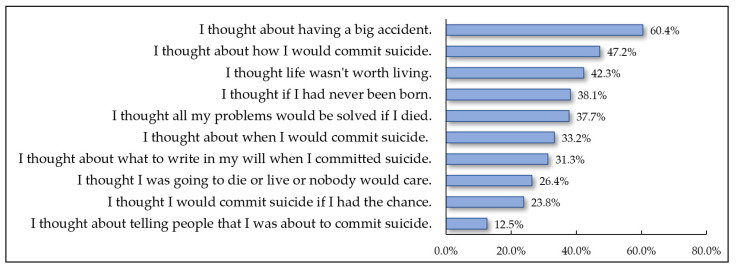
Response rate to the suicidal ideation question of participants.

**Figure 2 behavsci-13-00509-f002:**
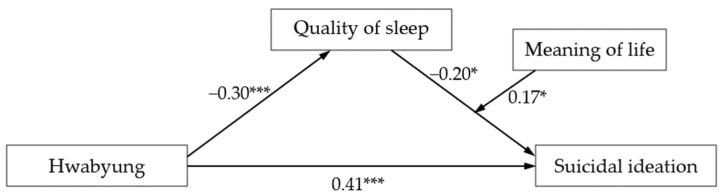
Research model with regression coefficients. * *p* < 0.05, *** *p* < 0.001.

**Figure 3 behavsci-13-00509-f003:**
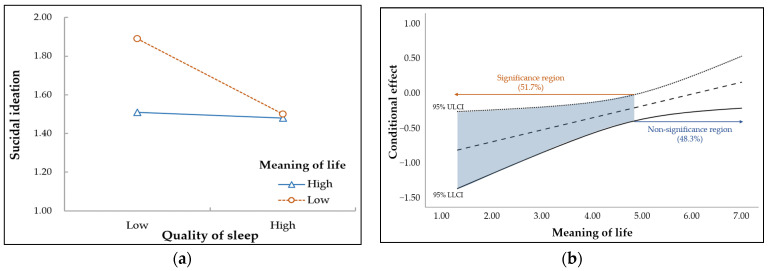
Moderated mediating effect of quality of sleep through meaning in life. (**a**) Conditional effect between quality of sleep and suicidal ideation according to the level of meaning in life. (**b**) Johnson–Neyman significance region of meaning in life.

**Table 1 behavsci-13-00509-t001:** Descriptive statistics and correlation coefficients (*N* = 265).

Variables	*M* ± *SD*	Hwabyung	Quality of Sleep	Meaning in Life
Hwabyung	2.46 ± 0.79			
Quality of sleep	2.99 ± 0.53	−0.46 ***		
Meaning in life	4.96 ± 0.96	−0.34 ***	0.19 **	
Suicidal ideation	1.61 ± 0.85	0.49 ***	−0.34 ***	−0.28 ***

** *p* < 0.01, *** *p* < 0.001.

**Table 2 behavsci-13-00509-t002:** Analysis of mediating effect of quality of sleep moderated by meaning in life on the relationship between Hwabyung and suicidal ideation (*N* = 265).

Model	Predictor	Reference Variable	*B*	*SE*	*p*
I	Hwabyung	Quality of sleep	−0.30	0.09	<0.001
II	Hwabyung	Suicidal Ideation	0.41	0.07	<0.001
	Quality of sleep	Suicidal Ideation	−0.20	0.10	0.041
	Meaning in life	Suicidal Ideation	−0.10	0.05	0.046
	Interaction ^a^	Suicidal Ideation	0.17	0.08	0.024

^a^ The interaction term was composed of the product of quality of sleep and meaning in life.

**Table 3 behavsci-13-00509-t003:** Result of bootstrapping analysis of conditional indirect effect.

Meaning in Life	Indirect Effect	Boot SE	Boot 95% CI
Low (M − 1SD)	0.11	0.04	[0.03, 0.20]
Middle (M)	0.06	0.03	[0.00, 0.13]
High (M + 1SD)	0.01	0.04	[−0.07, 0.09]

## Data Availability

The data used during the current study are available from the corresponding author on reasonable request.

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
