# Peer review of "Mediating Effect of Quality of Sleep Moderated by Meaning in Life on the Relationship between Hwabyung and Suicidal Ideation in Middle-Aged Korean Women"

_behavsci, 2023, doi:10.3390/bs13060509_

Round 1

Reviewer 1 Report

The study is well planned, but data processing needs improvement. In order to obtain the most accurate picture, two more important variables must be taken into account: the level of education and professional activity (if the participants in the study are housewives or professionally active). As a rule, a high-performance professional activity, with satisfaction, even if it alters the quality of sleep due to the engaging work, does not alter the psycho-emotional component but, on the contrary, motivates.

Reviewer 2 Report

This manuscript aims to investigate the role of quality of sleep and meaning of life in the process of Hwabyung symptoms affecting suicidal ideation in middle-aged Korean women. 

The manuscript presents some formal and substantial issues which should be addressed by the authors before considering publication

Introduction:

What do the authors mean by “suicide accidents”? please use a terminology proposed by Silverman et al., 2007. For example, the authors could use terms such as “died by suicide”, “suicide attempt” and, “suicidal ideation”.

In lines 30-31, what does the author mean by “Suicide cannot be studied in humans”?

In lines 41-42 please, better explain the sentence “Middle-aged women experience various physical and psychological symptoms due to hormonal decline and imbalance as they physiologically encounter menopause” 

in general, I would suggest that authors be more cautious when talking about "hwabyung" effects.

Please, rephrase this sentence “In summary, Hwabyung has a significant positive effect on suicide” it is useful not to confuse the reader with  "positive effect", you could use "a significant effect on".

Materials and methods

In this section, the authors could better explain how the study participants were recruited. 

For the Hwabyung Symptom Scale (HSS), it could be helpful if the authors reported some items.

Please change “Johnson-Nayman” to Johnson-Neyman.

Results

The results appear well reported, following APA rules

Discussion

In the discussions, it is not clear to me why the authors do not mention interventions focused on suicide risk reduction, as the data from this study also show a direct effect on suicidal ideation. 

In my opinion, the authors should emphasize the importance of suicide risk assessment in these women, as well as provide prevention programs.

The study would gain more value if in this section the authors focus both from the perspective of prevention and intervention. 

Overall, the present study is interesting and does not present serious flaws that impede the publication but it can be improved and should be double-checked for grammar and typos before being resubmitted.

I hope that my suggestions will be well received by the Authors, helping them to improve their manuscript.

it can be improved and should be double-checked for grammar and typos before being resubmitted

Round 2

Reviewer 1 Report

Accept the revised form of manuscript 

Author Response

Thank you again for your review.

Reviewer 2 Report

I thank the authors for following my suggestions. The manuscript has been improved. My only concern is about the sentence "Suicide is the end of one's own life, so it is impossible to study 'suicidal behavior' in real humans." 

I still don't understand what the authors mean by "real humans." In addition, suicidal behavior also refers to suicide attempts, so this sentence could be misinterpreted. There are several studies also that have studied post-mortem suicide through a method called "psychological autopsy."  

I suggest the authors revise the sentence or delete it.  

Author Response

Thank you again for your careful review. 
We have removed the controversial sentence as you suggested.